# Molecular Breeding of Water-Saving and Drought-Resistant Rice for Blast and Bacterial Blight Resistance

**DOI:** 10.3390/plants11192641

**Published:** 2022-10-08

**Authors:** Anning Zhang, Yi Liu, Feiming Wang, Deyan Kong, Junguo Bi, Fenyun Zhang, Xingxing Luo, Jiahong Wang, Guolan Liu, Lijun Luo, Xinqiao Yu

**Affiliations:** 1Key Laboratory of Grain Crop Genetic Resources Evaluation and Utilization, Ministry of Agriculture and Rural Affairs, Shanghai Agrobiological Gene Center, Shanghai 201106, China; 2State Key Laboratory of Biocatalysis and Enzyme Engineering, School of life Sciences, Hubei University, Wuhan 430062, China

**Keywords:** rice, WDR, blast, bacterial blight, *Pi2*, *xa5*, *Xa23*, MAS

## Abstract

Rice production is often affected by biotic and abiotic stressors. The breeding of resistant cultivars is a cost-cutting and environmentally friendly strategy to maintain a sustainable high production level. An elite water-saving and drought-resistant rice (WDR), Hanhui3, is susceptible to blast and bacterial blight (BB). This study was conducted to introgress three resistance genes (*Pi2*, *xa5,* and *Xa23*) for blast and BB into Hanhui3, using marker-assisted selection (MAS) for the foreground selection and a whole-genome single-nucleotide polymorphism (SNP) array for the background selection. As revealed by the whole-genome SNP array, the recurrent parent genome (RPG) recovery of the improved NIL was 94.2%. The resistance levels to blast and BB of the improved NIL and its derived hybrids were higher than that of the controls. In addition, the improved NIL and its derived hybrids retained the desired agronomic traits from Hanhui3, such as yield. The improved NIL could be useful to enhance resistance against biotic stressors and produce stable grain yields in *Oryza sativa* subspecies *indica* rice breeding programs.

## 1. Introduction

Rice (*Oryza sativa* L.) is a staple food source for more than half of the Earth’s population. Rice yield has doubled or even tripled in most rice-producing countries over the past three decades [1]. This is mainly due to the utilization of semi-dwarf breeding and the development of hybrid rice [2]. However, a rapid increase in the world’s population, coupled with a decrease in cultivable land, has exerted pressure on the production and productivity of rice [3]. Major barriers to high rice yield and the sustainability of rice production are the frequent occurrences of biotic stress, including various diseases caused by pathogens and abiotic stressors, such as drought, high temperature, and flood, due to global climate change. Therefore, it is necessary to develop new cultivars with resistance to both biotic and abiotic stressors.

Rice blast, caused by the fungus *Magnaporthe grisea*, is a devastating disease that leads to significant yield losses of up to 70–80% during an epidemic [4]. Each year, rice blast reportedly destroys rice yields that could have been used to feed an estimated 60 million people [5]. Currently, 84 blast resistance genes have been identified, and 24 of these genes (*Pb1*, *Pia*, *Pib*, *Pid2*, *Pid3*, *Pik*, *Pi54*, *Pik-m*, *Pik-p*, *Pish*, *Pit*, *Pita*, *Piz-t*, *Pi1*, *Pi2*, *Pi5*, *Pi2*, *pi21*, *Pi25*, *Pi36*, *Pi37*, *Pi56*, *Pi63*, and *PiCO39*) have been cloned and characterized (http://www.ricedata.cn/gene/gene_pi.htm (accessed on 29 August 2022)). A few of these blast resistance genes have already been incorporated into rice varieties that are now widely cultivated in many countries [6,7,8,9,10]. Among the blast resistance genes found in rice varieties grown in China, *Pi2* has been identified as one of the most effective genes, providing broad-spectrum resistance to rice blast [11,12,13]. The *Pi2* gene, existing in the *indica* rice line BL6, was introgressed from the wild species *Oryza minuta* [14]. A co-dominant sequence-tagged site (STS), marker PB9-1, which is linked to the *Pi2* gene, was developed for marker-assisted selection (MAS) [15]. With this in mind, the rice line BL6 was selected as the donor for MAS in this study.

BB, caused by *Xanthomonas oryzae* pv. *oryzae* (*Xoo*), is one of the most destructive rice diseases worldwide and causes significant yield reductions [16]. Presently, more than thirty-eight genes are known to confer resistance against BB, and seven of these genes (*Xa1*, *xa5*, *xa13*, *Xa21*, *Xa23*, *Xa26*, and *Xa27*) have been successfully cloned (http://www.ricedata.cn/gene/gene_xa.htm, accessed on 29 August 2022). Some resistance genes have been introgressed into the elite inbred lines and hybrids to develop BB-resistant rice varieties [17,18,19,20,21]. *Xa23*, identified from a wild species of rice, *Oryza rufipogon*, is a single completely dominant resistance gene that is effective at all growth stages [22]. The closely linked SSR marker RM206, which is 1.9 cM away from the *Xa23* locus, is available for MAS [23]. Therefore, CBB23 was used as the donor parent for MAS. CBB23 carries the *Xa23* gene and was derived from the BC_4_F_5_ progeny of the cross between an *O. rufipogon* accession (RBB16, the *Xa23* donor) and IR24, a recipient *indica* BB-susceptible variety [24].

Water-saving and drought-resistant rice (WDR) is a new type of rice variety that has an equally as good quality and yield potential as current paddy rice, yet it also has water-saving ability and drought resistance [25]. Hanhui3 is an elite WDR restorer line that was previously developed in our laboratory using conventional selection in extreme drought environments [26]. We developed several high-yielding, good-quality WDR hybrids using Hanhui3 as the male parent, such as ‘Hanyou73′ and ‘Hanyou113′. These hybrids are widely grown in arid areas in southern China. However, Hanhui3, ‘Hanyou73,’ and ‘Hanyou113′ are highly susceptible to blast and BB. Using pesticides to control diseases often increases farmers’ costs and raises concerns about food and environmental safety [27]. The urgent need to improve Hanhui3 for blast and BB resistance has merited the present study, which capitalizes on the availability of closely linked markers for the blast resistance gene *Pi2* and the BB resistance genes *xa5* and *Xa23*, and the application of inexpensive rice arrays for background selection.

## 2. Materials and Methods

### 2.1. Rice Materials

Hanhui3, an elite *indica* CMS rice restorer line with high drought resistance, good yield and good rice quality, but susceptible to blast and BB, was used as the recurrent parent. The donor parents were BL6, carrying the blast resistance gene *Pi2*, IRBB5, carrying the BB resistance gene *xa5*, and CBB23, carrying the BB resistance gene *Xa23*. Two CMS lines, Huhan7A and Huhan5A, have good yields and rice qualities but are susceptible to blast and BB; Huhan7A and Huhan5A were used as the maternal parents to produce the hybrid rice. 7A/Hanhui3 and 5A/Hanhui3, produced by Shanghai Tiangu Biotechnology Co., Ltd., (Shanghai, China), were used as the controls for the evaluation of the agronomic performance of the hybrid rice.

### 2.2. Marker-Assisted Selection of NIL for Blast and BB Resistance

Figure 1 illustrates the crossing scheme used to pyramid *Pi2*, *xa5,* and *Xa23* into Hanhui3. The overall breeding scheme consisted of a recurrent backcrossing procedure, including two crossings, two generations of backcrossing, and four generations of self-fertilization, combined with MAS in each generation and background selection in the last generation. The BC_1_F_1_, BC_2_F_1_, BC_2_F_2_, and BC_2_F_3_ lines were planted in drought conditions, and we conducted phenotype selection to select the drought-resistant plants. During the foreground and background selection of the BC_2_F_4_ lines, the improved NIL containing homozygous *Pi2*, *xa5*, and *Xa23* genes possessing maximum recurrent parent genome recovery was selected to cross with CMS to produce hybrids. The NIL and its derived hybrids from the CMS lines Huhan7A and Huhan5A were further evaluated for disease resistance and desired agricultural traits.

### 2.3. DNA Extraction, Markers, and Genotyping

Total genomic DNA was extracted from fresh rice leaves using the modified CTAB method [28]. Three linked markers, AP22, RM611, and RM206, were used to detect the presence of the *Pi2*, *xa5*, and *Xa23* genes, respectively (Table 1). Polymerase chain reactions were performed in a Bio-Rad PTC-200 Thermocycler (Bio-Rad, Hercules, CA, USA), and samples had a total volume of 20 μL, containing 20 ng of template DNA, 10 μM of each primer, 1.5 mM of MgCl_2_, 2 mM of dNTPs, and 1 U of Taq polymerase. Amplified products were separated on a 6% nondenaturing PAGE and visualized with silver staining [29].

In addition, the whole-genome SNP array RICE56K was used to analyze the genetic background similarities of the NILs compared with the recurrent parent Hanhui3. This SNP array contains 56,000 markers, from which we screened 33,782 high-quality SNPs distributed on the 12 chromosomes for background analysis. Total DNA was extracted from the leaves from 10 plants for all the NILs and the recurrent parent, Hanhui3. The analysis of genetic background similarity was conducted at Huazhi Rice Bio-tech Co., LTD (Changsha, China).

### 2.4. Screening for Blast Resistance

The recurrent parent, Hanhui3, and the NIL and its derived hybrids were screened for their reaction to blast infection under uniform conditions at a nursery in Jinggangshan, Jiangxi, China. The test materials were planted individually in a two-row plot with seven plants per row with a spacing of 25 × 15 cm. The rows were surrounded by the blast-susceptible cultivar ‘Yuanfengzao’ to ensure uniformity in the spreading of the blast infection. Twenty-five days after heading, the panicle blast of 100 panicles in each plot was investigated. The investigation and comprehensive evaluation of the rice blast were carried out according to the technical specification of the regional test for the identification and evaluation of blast resistance in rice varieties (NY/T 2646-2014).

### 2.5. Screening for BB Resistance

Hanhui3 and the NIL and its derived hybrids were grown in fields at the Shanghai Agrobiological Gene Center (Shanghai, China) and were screened for their resistance against BB via the artificial clip inoculation method. At the maximum tillering stage, the leaves of five plants from each plot were inoculated with ten mixed races (PXO341, PXO112, PXO339, PXO61, PXO17, PXO340, PXO280, PXO99, PXO86, and PXO146) of *Xoo* isolates with a bacterial suspension of approximately 10^8^ cells/mL. The *Xoo* isolates were obtained from IRRI. Fourteen days after inoculation, lesion lengths were measured in the 25 inoculated leaves (five leaves from five plants per plot). Resistance to BB was expressed by the lesion length measurements (resistant = <3 cm; moderately resistant = 3.1–5 cm; moderately susceptible = 5.1–12 cm; susceptible = 12.1–20 cm;) used by the standard evaluation system [32].

### 2.6. Evaluation of Major Agronomic Traits under Field Conditions

To evaluate the agronomic traits under normal conditions in the field, Hanhui3 and the NIL and its derived hybrids were planted in a four-row plot with seven plants per row with a spacing of 20 × 15 cm in a randomized complete block design with three replications. The field trials were conducted in Hefei, China, in the summer of 2020. After the rice was transplanted, intermittent irrigation was applied in the rice fields. The agronomic traits were measured according to the practices outlined by the standard evaluation system [32]. Data were recorded for five plants in the middle row to determine the following agronomic traits: days to 50% flowering, plant height, panicle length, number of tillers, number of grains per panicle, spikelet fertility, 1000-seed weight, and yield per plant.

To evaluate the agronomic traits of Hanhui3 and the NIL under drought conditions during the reproductive growth period, the identification of drought resistance was performed according to the reported method [33]. Both Hanhui3 and the NIL were planted in a four-row plot with nine plants per row, applying 20 × 23 cm spacing, in Zhuanghang Town, Shanghai, China. The treatments were repeated three times. Drought-stress treatment was conducted at stage II of young panicle differentiation. Five plants in the middle rows of the plots were measured for their agronomic traits.

### 2.7. Data Analysis

The recurrent parent genome (RPG) was calculated using the following formula: RPG (%) = (R + 1/2H) × 100/P, where P = the total number of parental polymorphic markers screened, R = the number of markers showing homozygosity for the recurrent parent allele, and H = the number of markers showing heterozygosity for the parental allele. One-way ANOVA and least significant difference (LSD) tests were performed using the SAS statistical software package (version 9.0; SAS Institute, Cary, NC, USA).

## 3. Results

### 3.1. Development of NIL with Pi2, xa5, and Xa23 Genes Using Marker-Assisted Selection

Marker-assisted backcross breeding was employed to transfer *Pi2*, *xa5,* and *Xa23* genes into the genetic background of Hanhui3. Two independent crossing programs were carried out for the development of the near-isogenic line (NIL), one using Hanhui3 as the female and BL6 as the male, and the other using IRBB5 as the female and CBB23 as the male. Two true F_1_ plants, which were confirmed by genetically linked markers, were crossed with each other to obtain F_1_ plants carrying *Pi2*, *xa5*, and *Xa23* genes. True F_1_ plants were identified by genetically linked markers and were backcrossed with Hanhui3 to produce BC_1_F_1_ seeds. The resulting BC_1_F_1_ plants were first checked for the presence of *Pi2*, *xa5*, and *Xa23* genes via genetically linked markers. Then, three positive plants possessing maximum similarity to the phenotype of Hanhui3 were used to generate BC_2_F_1_ seeds. Among 61 BC_2_F_1_ plants, 9 individuals containing *Pi2*, *xa5,* and *Xa23* genes and with a phenotype similar to Hanhui3 were self-crossed to produce BC_2_F_2_ seeds. In the BC_2_F_2_ population totaling 840 plants, 14 plants containing homozygous *Pi2*, *xa5,* and *Xa23* genes and a phenotype similar to Hanhui3 were selected to produce BC_2_F_3_ seeds (Appendix A). From the BC_2_F_3_ family lines, eight lines containing homozygous *Pi2*, *xa5,* and *Xa23* genes that had no phenotypic variation were selected to produce BC_2_F_4_ seeds.

In the BC_2_F_4_ plants, the genetic backgrounds of all the NILs were surveyed using the 56 K whole-genome SNP array. The recurrent parent genome (RPG) recoveries of the NILs were arranged from 88.6% to 94.2%. The NIL with the maximum RPG (94.2%) was identified through background selection and designated as R3-1. Haplotype maps were created to further ensure that the target loci were successfully transferred from the donor parents into the selected NIL and that the phenotype of the NIL was similar to that of the recurrent parents (Figure 2). These results suggest that through the pre-selection of target loci and phenotype, and the post-selection of genetic background, we could quickly, correctly, and economically obtain improved NILs with high RPG recoveries.

### 3.2. Blast and BB Resistance of the Improved NIL and Its Hybrids

The resistance levels to blast of the improved NIL and its derived hybrids were evaluated at the mature stage (Table 2). The recurrent parent, Hanhui3, was susceptible to rice blast with a disease score of 7.5. The improved NIL, R3-1, showed moderate resistance to rice blast with a disease score of 3.5. The hybrids 7A/Hanhui3 and 5A/Hanhui3 were moderately susceptible to rice blast with disease scores of 5 and 5.5. All the improved hybrids (7A/R3-1 and 5A/R3-1) showed moderate resistance to rice blast with disease scores of 3 and 2.5. These results demonstrate that the heterozygous and homozygous *Pi2* gene in the NIL and its three-line hybrids improved their disease resistance to rice blast.

The improved NIL and its derived hybrids were evaluated for their resistance levels to BB at the tillering stage (Table 2). The recurrent parent, Hanhui3, was observed to be moderately susceptible to BB, with a lesion length of 15.22 cm. The improved NIL showed resistance to BB, with a lesion length of 1.51 cm. The hybrids 7A/Hanhui3 and 5A/Hanhui3 were moderately susceptible to BB, with lesion lengths of 8.82–9.03 cm, while the improved hybrids (7A/R3-1 and 5A/R3-1) had some resistance to BB, with lesion lengths of 2.09–2.28 cm. These results demonstrate that the NIL and its derived three-line hybrids showed improved disease resistance to BB.

### 3.3. The Performance of Agronomic Traits of the NIL and Its Hybrids

For the purpose of surveying whether the other agronomic traits of the improved NIL and its derived hybrids were identical to those of the recurrent parent and its hybrids, we tested eight agronomic traits of the improved NIL and the hybrids (Table 3). The improved NIL displayed no significant difference compared to Hanhui3 in traits such as days to 50% flowering, plant height, panicle length, number of tillers per plant, spikelet fertility, 1000-seed weight, and yield per plant. However, the number of grains per panicle of R3-1 was higher than that of Hanhui3.

Hybrids derived from the improved NIL showed no significant differences compared with the hybrids derived from the recurrent parent, Hanhui3, in traits such as days to 50% flowering, plant height, no. of tillers, no. of grains per panicle, 1000-seed weight, and yield per plants. However, the panicle length of 7A/R3-1 was 25.72 cm, which was longer than that of 7A/Hanhui3, and the differences were significant. The spikelet fertility of 7A/R3-1 was 88.19%, which was significantly higher than that of 7A/Hanhui3. The results indicated that there were no significant differences in major agronomic traits under drought yield conditions in the improved NIL and its derived hybrids.

To evaluate the drought tolerance phenotype of Hanhui3 and the NIL, we also measured eight agronomic traits under drought conditions (Table 4). The results of the drought resistance identification show that there were no significant differences in traits between Hanhui3 and the NIL under drought conditions. These results indicate that the NIL retained the drought resistance characteristic of the recurrent parent.

## 4. Discussion

Blast and bacteria blight are detrimental to the production and yield of hybrid rice in southern China; however, most of the available hybrids do not have resistance to either disease [34]. Hanhui3, an elite restorer line, possesses a broad spectrum of fertility restoration traits, with high drought resistance, and good grain quality and lodging resistance. The hybrid Hanyou73, renowned for its unique water-saving and drought-resistant characteristics, was produced for commercial cultivation in 2011, using Hanhui3 as the male parent [35]. Despite its water-saving and drought-resistant traits and its popularity among farmers, Hanhui3 and its hybrids are still problematic, as they are also highly susceptible to blast and BB. Because drought tolerance is a complex quantitative trait, there is no available drought-related molecular marker; thus, introgressing drought tolerance traits into lowland rice with MAS is not possible [25]. In order to combine the disease-resistant genes *Pi2, xa5,* and *Xa23* with drought-tolerant traits, we used the following breeding strategy: 1. the introgression of disease-resistant genes into a water-saving and drought-resistant rice background; 2. the use of conventional breeding to directly select for yield under drought screening to maintain drought resistance; 3. the application of MAS for the foreground selection of disease resistance genes; and 4. the utilization of a 56 K rice array for the background selection to obtain the improved NIL with the highest RPG. In the present study, we successfully transferred the disease resistance genes *Pi2, xa5,* and *Xa23* while maintaining the recurrent parent’s drought resistance. Our results suggested that the integration of conventional breeding, MAS, and rice 56 K array technology in backcross breeding ensures accuracy when transferring the desired genes to the recipient parent and reduces the number of backcrossing cycles, compared with conventional backcross breeding when seeking to improve multiple traits.

‘Linkage Drag’ is a common phenomenon in disease breeding [36]. The donor parents, BL6, IRBB5, and CBB23, carry the blast resistance genes *Pi2*, xa5, and *Xa23*, but were relatively inferior in water-saving and drought-resistant traits when compared with the recurrent parent, Hanhui3. Therefore, recovering the water-saving and drought-resistant traits present in Hanhui3, while simultaneously introgressing the disease resistance of the donor parents, was particularly challenging in this breeding strategy. The major focus of this experiment was to transfer the target genes with a minimum number of undesirable donor segments to create the improved NILs. Hence, the use of MAS for target genes, followed by the phenotypic selection of plants with a similar agronomic performance to Hanhui3, drought screening, and background selection using a 56 K array for the BC_2_F_4_ lines increased the RPG of the improved NILs. The results showed that the maximum RPG of the improved NILs was 94.2%, which is higher than the theoretical values. It is apparent from the haplotype maps that the chromosome is almost recovered for the Hanhui3 segment, except for chromosomes 5, 6, and 11. Linkage drag was obvious in chromosomes 5, 6 and, 11 because of the MAS of target genes. In future studies, the breakage of the observed linkage drag would depend on a larger segregating population and the background selection of earlier generations.

The disease resistance of heterogeneous genotypes is important for hybrid improvement. Because some disease resistance genes display partial dominant or recessive resistance, their heterozygotes show moderate resistance or can even be susceptible to disease [37,38,39]. If these genes are transferred to a hybrid parent, they have to be simultaneously introgressed into two parents to guarantee that the resulting hybrids also possess disease resistance. In this study, the improved NIL with the transferred disease resistance genes *Pi2, xa5,* and *Xa23* showed enhanced resistance to blast and BB compared with the controls. When this NIL was crossed with the CMS lines 7A and 5A, the hybrids (heterogeneous genotypes with resistance loci) showed a similar disease resistance to the homologous NIL. In accordance with previous studies [15,22], these results indicate that genes have a great potential for future hybrid improvement. Interestingly, the lesion length of the BB of the recurrent parent Hanhui3 was significantly longer than those of the derived hybrids. However, the lesion length of the improved NIL was even shorter than those of the derived hybrids. These results are likely caused by the recessive gene *xa5*.

In addition to rice disease resistance, we focused on the yields of the improved NIL and its derived hybrids in drought conditions. The main agronomic traits of the improved NIL and its derived hybrids were similar or superior to those of the controls. The spikelet fertility of the improved NIL was even higher than that of Hanhui3, indicating that there was no yield penalty due to the pyramiding of the resistance genes. In addition, these results may be attributed to the strict phenotype selection of desirable traits in every backcrossed generation, combined with the background selection conducted in the last generation. Furthermore, this is consistent with results from simulation studies showing that using foreground and background selection could lessen two or three BC generations to complete backcross agronomical traits compared with conventional backcrossing [40].

The hybrid 5A/R3-1 exhibited blast and BB resistance, as well as the highest yield among the investigated hybrids. We affirm that this hybrid could be used for rice production in the upland regions of southern China. Furthermore, when compared with the donors, IRBB5, CBB23, and BL6, the improved NIL not only demonstrated blast and BB resistance, but also high-yield characteristics. Therefore, the improved NIL could be a suitable donor for future breeding studies. The improved NIL would be useful for enhancing the effective resistance to multiple biotic stressors and producing stable grain yield in *indica* rice.

## Figures and Tables

**Figure 1 plants-11-02641-f001:**
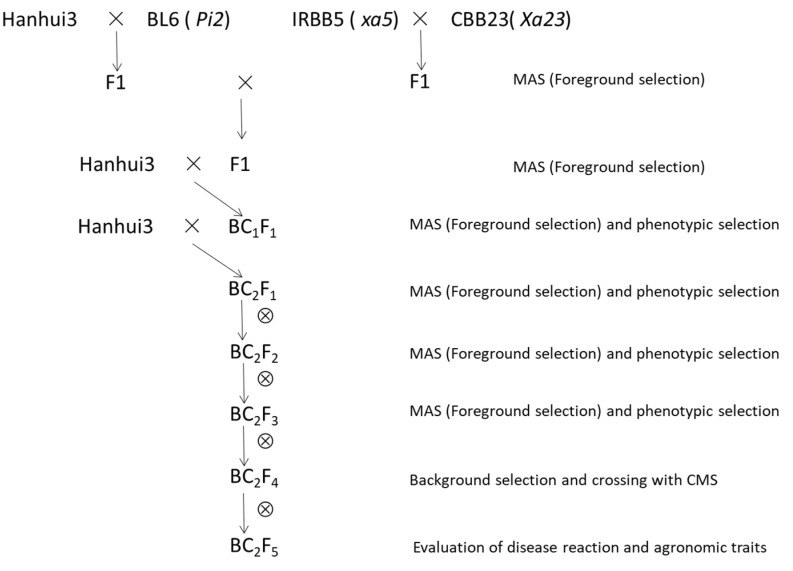
Schematic diagram for the development of NILs with homozygous *Pi2*, *xa5*, and *Xa23* loci in the genetic background of Hanhui3. MAS, marker-assisted selection. CMS, cytoplasmic male sterility.

**Figure 2 plants-11-02641-f002:**
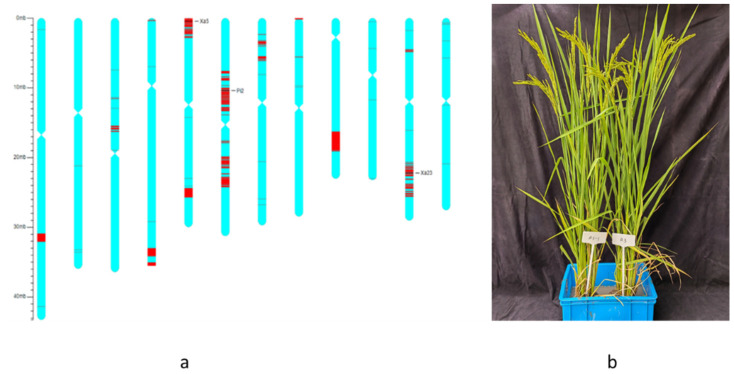
Genetic background assay and phenotype of the improved NIL. (**a**) The black lines indicate the positions of three target genes, *xa5* on chromosome 5, *Pi2* on chromosome 6, and *Xa23* on chromosome 11. The red lines indicate the SNP loci with homozygous genotypes where genomic fragments of the donor parents were introgressed. The green lines indicate the SNP loci with the same genotypes as the recurrent parent Hanhui3. (**b**) The phenotype of the improved NIL and recurrent parent. Left is NIL. Right is recurrent parent Hanhui3.

**Table 1 plants-11-02641-t001:** Linked markers for foreground selection of *Pi2*, *xa5*, and *Xa23* genes.

R Genes	Chr. No.	Marker Name	Primer Sequences (5′-3′)		Exp. Size (bp)	Band Type	Reference
			Forward Primer (F)	Reverse Primer (R)			
*Pi2*	6	AP22	GTGCATGAGTCCAGCTCAAA	GTGTACTCCCATGGCTGCTC	143	Co-dominant	[30]
*xa5*	5	RM611	CAACAAGATGGCCTCTTACC	TACAAACAAACAGCTTGTGC	213	Co-dominant	[31]
*Xa23*	11	RM206	CCCATGCGTTTAACTATTCT	CGTTCCATCGATCCGTATGG	147	Co-dominant	[23]

**Table 2 plants-11-02641-t002:** Blast and bacterial blight resistance of the improved NIL and its derived hybrids.

Rice Line	Resistance Genes Genotype #			Reaction against Blast *		Reaction against BB *	
	*Pi2*	*xa5*	*Xa23*	Score	Disease Reaction	Lesion Length	Disease Reaction
Hanhui3	--	--	--	7.5	S	15.22 ± 3.24	S
R3-1	++	++	++	3.5	MR	1.51 ± 0.05	R
7A/Hanhui3	--	--	--	5	MS	8.82 ± 2.18	MS
7A/R3-1	+-	+-	+-	3	MR	2.28 ± 0.37	R
5A/Hanhui3	--	--	--	5.5	MS	9.03 ± 2.31	MS
5A/R3-1	+-	+-	+-	2.5	MR	2.09 ± 1.21	R

# ++, homozygous genotypes with the donor parents; --, homozygous genotypes with the recurrent parents; +-, heterozygous genotypes. * R, resistant; MR, moderately resistant; MS, moderately susceptible; S, susceptible; HS, highly susceptible.

**Table 3 plants-11-02641-t003:** The yields and agronomic performances of NIL and its derived hybrids under normal field conditions.

Rice Line	Days to 50% Flowering	Plant Height (cm)	Panicle Length (cm)	No. of Tillers	No. of Grains per Panicle	Spikelet Fertility (%)	1000-Seed Weight (g)	Yield per Plant (g)
Hanhui3	102.2 ± 3.5	117.45 ± 4.82	21.32 ± 1.83	6.65 ± 0.64	186.54 ± 8.71	87.34 ± 6.15	26.85 ± 0.44	29.11 ± 5.62
R3-1	102.7 ± 1.2	115.13 ± 3.61	22.04 ± 0.87	6.75 ± 2.21	201.09 ± 7.81 *	85.27 ± 1.78	26.29 ± 0.31	28.38 ± 3.61
7A/Hanhui3	88.0 ± 2.1	123.50 ± 6.14	24.15 ± 0.54	7.54 ± 1.04	177.32 ± 21.36	84.52 ± 2.84	29.33 ± 0.88	34.21 ± 3.89
7A/R3-1	88.3 ± 1.8	122.77 ± 5.57	25.72 ± 0.21 *	7.44 ± 0.69	165.51 ± 10.37	88.19 ± 1.85 *	30.68 ± 1.75	37.19 ± 2.41
5A/Hanhui3	93.2 ± 4.3	116.50 ± 2.64	23.52 ± 0.85	9.42 ± 0.53	182.13 ± 7.62	79.42 ± 1.64	26.41 ± 1.58	40.92 ± 6.48
5A/R3-1	91.4 ± 3.1	116.35 ± 5.71	24.74 ± 0.36	9.82 ± 0.82	185.12 ± 11.46	85.12 ± 3.68	25.98 ± 1.23	42.63 ± 5.63

The * indicates a significant difference compared with the performance of controls at * *p* < 0.05.

**Table 4 plants-11-02641-t004:** The yields and agronomic performances of Hanhui3 and NIL under drought conditions.

Rice Line	Days to 50% Flowering	Plant Height (cm)	Panicle Length (cm)	No. of Tillers	No. of Grains per Panicle	Spikelet Fertility (%)	1000-Seed Weight (g)	Yield per Plant (g)
Hanhui3	105.3 ± 3.21	87.06 ± 2.46	17.92 ± 1.54	5.33 ± 2.57	89.22 ± 8.52	69.30 ± 4.48	23.28 ± 1.76	15.06 ± 1.22
R3-1	105.9 ± 2.42	86.50 ± 4.23	19.76 ± 2.58	5.00 ± 1.52	100.58 ± 11.72	78.97 ± 5.72	24.52 ± 1.51	14.95 ± 0.98

## Data Availability

The datasets presented in the study are either included in the article or in the Appendix A; further inquiries can be directed to the corresponding authors.

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
