# Peer review of "Molecular Breeding of Water-Saving and Drought-Resistant Rice for Blast and Bacterial Blight Resistance"

_plants, 2022, doi:10.3390/plants11192641_

Round 1

Reviewer 1 Report

This MS present introduced resistance genes for rice blast and bacterial blight, which were insufficient in breeding excellent varieties suitable for climate crisis. As a result, it comes a hybrid variety that has water-saving and drought-resistance with disease resistant. However, the content of this article appears to be different from the area of this journal. This article may suitable to 'agriculture' journal, like that.

Also, some sentences have different explanations than the results. Paragraph 2 on page 7 :

'However, the panicle length ~ longer than the 1,000-seed weight'.

Line 161 ANVOA -> ANOVA

Author Response

In this work, we reported the combination of MAS and high throughput genotyping of rice germplasm for the rapid introgression of disease resistance genes in water saving and drought-resistant genomic background. So we think this research is suitable for publication in the journal of Plants, Special Issue ‘Opportunities and Challenges in Plant Germplasm’.

Thank you for pointing out two spelling mistakes in the article, and we corrected them in the revised version.

Reviewer 2 Report

The manuscript describes the combination of MAS and high throughput genotyping of rice germplasm for the rapid introgression of disease resistance genes with minimal perturbation of the drought-resistant genomic background. 

The introduction makes a clear description of the problem with breeding elite rice varieties that combine disease resistance and drought-tolerance. They cite appropriate references and clearly state their objective.

The methods are well described. I would suggest making an emphasis that the field trials and overall breeding and selection stages were done under water stress conditions. On line 146 it states "To evaluate the agronomic traits under normal conditions in the field (...)" which describes the conditions for agronomic traits measurement but not the selection and breeding conditions. The lack of a description of selection and breeding conditions undermines the conclusion that the new varieties maintain the water-saving and drought-resistance trait of the Hanhui3 parent.  Yet, this is described in the Discussion section. It would also be useful to describe the  software and thresolds  used to estimate the genomic background after the 56K SNP genotyping. Dinally, I would suggest adding the total sample size used for the measurement of disease resistance phenotypes. Is it correct that only 5 plants were used for measurement of agronomic traits?

The results are clearly described and persuasive. Table 3 should include a 95% confidence interval or standard error of the measurements as in Table 2. Insufficient sample size would make it difficult to find significant differences even if there was one.

The discussion is well written and clearly makes a case for the effectiveness of the breeding protocol and the future usefulness of the new varieties. I was surprised by the lack of essays testing drought-tolerance until I read lines 257 and 258 "(...) 2. using conventional breeding to directly select for yield under drought screening to maintain drought tolerance (...)". This description should be clearly stated in the methods section under "Marker-assisted selection of NIL for blast and BB resistance". 

It could be useful to include an assessment of the feasability of applying  this breeding and screening protocol on other crops. 

Author Response

Thank you all for your excellent suggestions. In the revised version, we described the process of breeding in detail. In order to retain the drought resistance character, the BC1F1, BC2F1, BC2F2 and BC2F3 lines were planted in drought condition, and we conducted phenotype selection to select drought resistant plants. In the supplement materials, the agronomic traits of the NIL under drought conditions were added, and the results showed that compared with control, the NIL also had drought resistance.

S Table 1 The yield and agronomic performance of NIL under drought yield conditions.

Rice line

Days to 50% flowering

Plant height (cm)

Panicle length (cm)

No. of tillers

No. of grains per panicle

Spikelet fertility

(%)

1,000-seed weight

(g)

Yield per plant

(g)

Hanhui3

105.3

87.06

17.92

5.33

89.22

69.30

23.28

15.06

R3-1

105.9

86.50

19.76

5.00

100.58

78.97

24.52

14.95

The * indicates a significant difference from the performance of controls at *P<0.05.

In the Materials and Methods part of the article, we added the sample sized used for the measurement of panicle blast. According to the technical specification for identification and evaluation of blast resistance in rice variety regional test (NY/T 2646-2014), the panicle blast of 100 panicles in each plot was investigated. Fourteen days after BB inoculation, the 25 inoculated leaves of each plot were measured of lesion length.

In the Results parts of the article, we added the standard error in Table3.

Reviewer 3 Report

For multiple target gene introgression by marker-assisted selection, linkage drags have always been a research issue for plant breeders. The authors showed that by combinations of phenotypic selection and genome-wide SNP array screening, the large RPG can be retained in advanced BC progenies, ranging from 88-94% which is higher than expected BC2 without background selection. However, there were still target and non-target linkage drags left on the RPG. The agronomic evaluation of advance BC progenies did not show deleterious impact from such linkage drags. I also suggested the authors present the amount of linkage drags on the target and non-target genomic regions. Please suggest in the discussion innovative approaches to effectively prevent and remove linkage drags in backcross breeding. 

Fig 2 on SSLP can be removed to Supplement Fig as they did not give any new knowledge about MAS. 

L261: The statement about retaining drought tolerance in the NIL need to have supporting data. If no supporting data, the authors must tone down the statement to expectation and propose what should be done in the future experiment.

L292-3: xa5 is a recessive R gene, why did hybrids with resistant NIL show better resistance to BB. Could Xa 23 have a stronger impact against BB than xa5? Please discuss this further.  

Author Response

Thank you all for your excellent suggestions. In the revised version, we described the process of breeding in detail. In order to retain the drought resistance character, we had carried out several drought resistance screening to select drought resistance plants. In the supplement materials, the agronomic traits of the NIL under drought conditions were added in S Tablel1. The results showed that the NIL also had drought resistance.

S Table 1 The yield and agronomic performance of NIL under drought yield conditions.

Rice line

Days to 50% flowering

Plant height (cm)

Panicle length (cm)

No. of tillers

No. of grains per panicle

Spikelet fertility

(%)

1,000-seed weight

(g)

Yield per plant

(g)

Hanhui3

105.3

87.06

17.92

5.33

89.22

69.30

23.28

15.06

R3-1

105.9

86.50

19.76

5.00

100.58

78.97

24.52

14.95

The * indicates a significant difference from the performance of controls at *P<0.05.

In the revised version, Fig. 2 was removed to Supplement Fig.1.

In the discussion part of the article, we discuss the result of BB resistance levels. Our results demonstrate that the NIL and its derived three-line hybrids showed resistance to BB disease resistance.But, the lesion length of BB of recurrent parent Hanhui3 was significantly longer than that of the derived hybrids. However, the lesion length of the improved NIL was even shorter than that of the derived hybrids. That results are likely caused by recessive gene xa5.
